# TradeFM: A Generative Foundation Model for
# Trade-flow and Market Microstructure

**Srijan Sood** [*]  **Maxime Kawawa-Beaudan** [*]  **Daniel Borrajo**  **Manuela Veloso**

J.P. Morgan AI Research, New York, NY, USA

## Abstract

Foundation models have transformed domains from language to genomics by learning general-purpose representations from large-scale, heterogeneous data. We introduce TradeFM, a 524M-parameter generative Transformer that brings this paradigm to market microstructure, learning directly from 10.7 billion trade messages across >9,000 US equities. To enable cross-asset generalization, we develop scale-invariant features and a universal tokenization scheme that map the heterogeneous, multi-modal event stream of order flow into a unified discrete sequence – eliminating asset-specific calibration. Integrated with a deterministic market simulator, TradeFM-generated rollouts reproduce key stylized facts of financial returns, including heavy tails, volatility clustering, and absence of return autocorrelation. Quantitatively, TradeFM achieves 2–3× lower distributional error than Compound Hawkes baselines and transfers zero-shot to geographically out-of-distribution APAC markets with substantial perplexity overlap. Together, these results suggest that scale-invariant trade representations capture transferable structure in market microstructure, opening a path toward synthetic data generation, stress testing, and learning-based trading agents.

## 1. Introduction

Foundation models for structured data have advanced rapidly along two axes: tabular pre-training (Hollmann et al., 2025) and time-series forecasting at scale (Ansari et al., 2024; Das et al., 2024; Woo et al., 2024). Both paradigms assume regularly-sampled observations with continuous or scalar-categorical targets – assumptions we relax in this work. Financial markets generate the canonical structured-data stream that violates these assumptions: order events arrive irregularly at sub-millisecond resolution, mixing continuous magnitudes (price, volume, inter-arrival time) with categorical fields (action, side) at heavy-tailed empirical distributions (Cont, 2001).

Despite this complexity, there is strong evidence that price formation follows *universal* principles across markets. Sirignano & Cont (2021) showed that a single deep learning model trained on pooled multi-stock data outperforms asset-specific models, even for held-out stocks. The atomic unit of financial markets is the individual order or trade event; modeling at this granularity yields native event-stream generation and subsumes upstream representations (L2 LOB depth snapshots, L1 top-of-book time series) by natural reconstruction. Recent foundation-scale efforts on order-flow data cover at most ∼500 assets and consume reconstructed limit-order-book (Level-2) snapshots (Li et al., 2025) or aggregated K-line OHLC bars (Shi et al., 2025). TradeFM is the first generative foundation model trained directly on Level-3 (L3) message-level order flow at the breadth of US equities (>9,000), preserving the L3 granularity that participants and exchanges actually observe and enabling zero-shot perplexity transfer across geographic markets. This finding motivates a natural question: *can a single foundation model learn a general-purpose representation of market mechanics from raw, multi-asset order flow?*

Our contributions are fourfold:

1. **TradeFM**: A large-scale generative foundation model for market microstructure that learns unified trade-flow dynamics from billions of trade messages across the breadth of the US equity market.

2. **Learning from Partial Observations**: Unlike prior deep learning approaches that require the full limit order book as input (Table 1), TradeFM learns from a partially observed market state – the event stream available to any single participant – demonstrating that a foundation model for microstructure does not require privileged access to the full order book.

---

[*]Equal contribution. Correspondence to: Srijan Sood <srijan.sood@jpmorgan.com>.
*ICML 2026 Workshop on Foundation Models for Structured Data (FMSD)*, Seoul, South Korea, 2026. Copyright 2026 by the author(s). Extended version: arXiv:2602.23784.

| | DeepLOB | MaRS | LOBS5 | Kronos | **TradeFM** |
|---|---|---|---|---|---|
| Input | L2 snap. | L2 snap. | L2 + L3 | K-line | **L3 msg.** |
| Assets | 5 | 500 | 2 | ~1,000s | **>9,000** |
| Params | 60K | ~1B | 6.3M | 499M | **524M** |
| Zero-shot | No | No | No | Yes | **Yes** |

*Table 1.* **Comparison with related microstructure models** (Zhang et al., 2019; Li et al., 2025; Nagy et al., 2023; Shi et al., 2025). TradeFM uniquely combines the most parsimonious input – L3 trade messages alone, without book reconstruction or aggregation – with multi-thousand-asset coverage and zero-shot geographic transfer.

3. **Scale-Invariant Representation and Tokenization**: An end-to-end methodology that transforms raw, heterogeneous trade data into a unified discrete sequence via scale-invariant features and a universal tokenization scheme, enabling a single model to generalize across diverse assets and liquidity regimes without asset-specific calibration.

4. **Closed-Loop Market Simulation**: Integration of the pre-trained model with a deterministic market simulator, creating an environment for evaluating realism via stylized-fact reproduction, with forecasting, world modeling, market-impact analysis, and learning-based agent training as intended downstream applications.

Prior models require either full LOB reconstruction (DeepLOB, MaRS, LOBS5) or aggregate OHLC bars (Kronos); TradeFM operates on raw trade messages alone – the lowest-infrastructure data assumption a participant or institution can satisfy. Section 2 describes the method, Section 3 reports results, Section 4 outlines future directions; preliminaries (microstructure primer, related work, FMSD positioning) are in Appendix A.

## 2. Method

### 2.1. Trade Events as a Structured Sequence

A trade message *for a given asset* at time $t$ is the tuple $(\Delta t_t, \delta p_t, v_t, a_t, s_t)$: inter-arrival time $\Delta t_t$, normalized price depth $\delta p_t$, volume $v_t$, action type $a_t \in \{\text{add}, \text{cancel}\}$, and side $s_t \in \{\text{buy}, \text{sell}\}$.[1] Conceptually, each event is a row in an irregularly-sampled, mixed-type structured table whose columns have very different cardinalities and tail behavior. The mid-price is unobserved – we estimate $\hat{p}_t^{\text{mid}}$ from execution prices via an exponentially-weighted volume-weighted-average (EW-VWAP, Appendix B) – so depth is computed as a unit-less ratio $\delta p_t = (p_t^{\text{order}} - \hat{p}_t^{\text{mid}})/\hat{p}_t^{\text{mid}}$, comparable across assets quoted at very different price levels. Volume is log-transformed; both transformations make features scale-invariant across the 9,000+ asset universe (Sirignano & Cont,

---

[1]Terminology: *trade*, *event*, and *order* refer interchangeably to an order-flow message (submission or cancellation); a completed trade is a *fill* (execution).

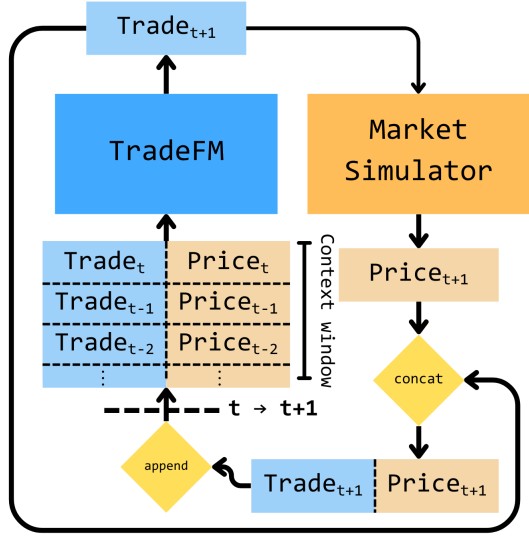

*Figure 1.* **Closed-loop simulation architecture.** TradeFM predicts a trade, the Market Simulator executes it, and the updated market state is fed back to the model.

2021). TradeFM trains under partial observability: the model sees order events alone, never the full book, making the modeling regime reproducible from any sufficiently diverse trade-message corpus – exchange or large participant – without book-reconstruction infrastructure. The asset identifier is not a feature: a single model trained on the pooled multi-asset corpus generalizes to assets it has never seen during training, as demonstrated by zero-shot transfer to Chinese and Japanese markets (Figure 2). Because the underlying features $(\Delta t, \delta p, v, a, s)$ are fundamental to any continuous-trading market, the asset-agnostic representation extends in principle across asset classes (FX, fixed income, crypto). Full data sources, mid-price estimator derivation, and scale-invariant feature construction are in Appendix B.2, Appendix B.3, and Appendix B.4.

### 2.2. Mixed-Radix Tokenization

Standard Transformers operate on univariate sequences with a single token vocabulary; our trade event is a multi-feature tuple of mixed continuous and categorical values. We tokenize in two steps. *(1) Discretization*: each continuous feature is binned into 16 buckets – equal-frequency (quantile) for $\delta p$, equal-width on the log-transformed scale for $\Delta t$ and $v$ (heavy-tailed); discrete features $(a, s)$ are 2-way each. The tokenizer is calibrated on the first 30 days of training data with the top 1% (and bottom 1% for signed features) clipped before binning (Appendix B Algorithm 1). *(2) Mixed-radix encoding*: per-feature bin indices combine into a single composite token over a flat $|\mathcal{V}| = 16{,}384$ joint vocabulary via mixed-radix product binning ($16 \times 16 \times 16 \times 2 \times 2$). The model predicts one token per event from a softmax of width $|\mathcal{V}|$ – one prediction, one cross-entropy loss, no per-feature heads or sub-token autoregressive unrolls. Three contextual features –

liquidity tier ($n_l$=3), price-level-change bin ($n_{\Delta p}$=32), and a market-vs-participant indicator – pass through as separate input embeddings, not predicted (composite-token equation and worked example in Appendix B; bin-edges visualization in Figure 4).

A flat $|\mathcal{V}|$-way *output softmax* is ∼16.8M parameters – about 3% of TradeFM's 524M total – with a single matmul per event. Per-feature heads or sub-token autoregressive decoding either reintroduce conditional dependencies or pay $O(K)$ rollout cost (where $K$ is the per-event feature count), neither of which buys capacity at this vocabulary size. Patch-based time-series tokenizers (Ansari et al., 2024; Woo et al., 2024) encode regularly-sampled scalar targets and treat time as a sampling rate; in TradeFM, time is a feature ($\Delta t_t$), and there is no temporal patching.

### 2.3. Architecture and Pre-training

TradeFM is a Llama-family decoder-only Transformer (Touvron et al., 2023; Vaswani et al., 2017) with 524M parameters (32 layers, $d_{\text{model}} = 1024$, $d_{\text{ff}} = 4096$, 32 attention heads with 8 KV heads, RoPE positional encoding, 1,024-token context). The training corpus spans 368 trading days (Feb 2024 – Sep 2025) across >9K US equities, with 10.7B tokens for training (pre-Jan 2025) and 8.7B tokens for evaluation. We follow Chinchilla-optimal allocation (Hoffmann et al., 2022) ( 20 tokens/parameter) and train for 4 epochs (Muennighoff et al., 2023) with AdamW (peak LR $5\times10^{-5}$, linear schedule, 500 warm-up steps, effective batch 4,032, fp16 on 3×A100-80GB; full hyperparameters and tabular input-embedding details in Appendix B).

### 2.4. Closed-Loop Simulation

For deployment and audit, each step must be interpretable: what trade did the model emit, and what market state resulted? We pair TradeFM with a deterministic limit-order-book simulator implementing price-time-priority matching (Figure 1). At each step, multinomial sampling with a repetition penalty (1.2) draws a token from the model; the simulator decodes it into an order action, updates the book, and feeds the resulting state back into the next-token context (Nasdaq Listing Center, 2024). This separates learned order-flow dynamics (the model) from market-mechanism logic (the simulator), so the simulator's structural invariants (queue priority, no negative depth) are imposed externally rather than learned (full pseudocode in Algorithms 2 and 3; ZI and Hawkes baselines in Appendix B.10). The recursive loop generates long, dynamic sequences of market activity and enables the study of second-order effects like price impact, as the model's own predictions influence the market state that conditions its future predictions.

## 3. Results

**Evaluation Hierarchy.** We evaluate TradeFM bottom-up across the data hierarchy: (1) one-step event-forecast *perplexity* on held-out US data and across regimes – geographic (APAC markets) and temporal (Jan – Sep 2025 held-out, plus 2020 → 2025 forward-test on the ITCH variant); (2) *stylized-fact* reproduction over log-return marginals (heavy tails, volatility clustering, return autocorrelation, aggregational Gaussianity); and (3) *distributional fidelity* of emergent microstructure features – spreads, order-book imbalance, depth, volumes – using the Kolmogorov–Smirnov (KS) and Wasserstein ($W_1$) distance suite established by LOB-Bench (Nagy et al., 2025).

**Protocol.** For experiments (2) and (3), we sample 10 closed-loop rollouts of 1,024 events each for 9 assets across 3 liquidity tiers in each of 9 held-out months – **810** rollouts in total – conditioned on 1,024 real historical events. The three tiers are the low/medium/high average-daily-volume (ADV) bins used for the $i_l$ conditioning feature (Appendix B.6). The temporal hold-out (Jan 2025 onward) provides contamination protection: the tokenizer is calibrated only on the first 30 days of training data, so test-set statistics never inform binning (Appendix F). Baselines are zero-intelligence (ZI) traders (Gode & Sunder, 1993; Farmer et al., 2005) and a four-dimensional Compound Hawkes process (Bacry et al., 2015; Jain et al., 2024) with GMM-fit price depths and exponential volumes – canonical statistical anchors for partial-observability order-flow modelling. Modern neural generators across the GAN, VAE, and diffusion families (e.g. TimeGAN (Yoon et al., 2019), CTGAN (Xu et al., 2019)) typically target price or return series rather than event-level order flow; adapting them to asynchronous L3 event streams with mixed continuous-discrete types requires reformulation through our event-stream tokenization, which we leave to follow-on work. We mean-variance normalize distributions before computing $W_1$ to make the metric comparable across quantities.

### 3.1. Perplexity

**Geographic Transfer.** Direct evidence that TradeFM has learned transferable structure rather than US-specific quirks is **cross-mechanism transfer** (Figure 2). Held-out Chinese and Japanese markets (Jan 2025) have substantial microstructural differences from the US training distribution: Japan uses an Itayose batch-auction open and close (vs continuous trading), China imposes ±10% daily price limits (vs none in the US), and Asian bid–ask spreads are typically 3–10 bps (vs 1–2 bps in the US). On Chinese markets, per-asset perplexity tracks US baselines closely (median ∼18 vs US ∼17, a ∼6% shift); on Japanese markets, perplexity shifts to about 2× US (median ∼34 vs ∼17), still showing substantial distributional overlap with US held-out perplexity.

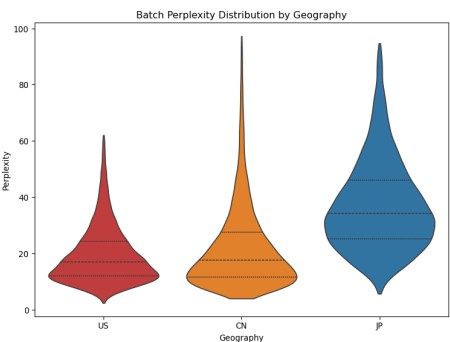

*Figure 2.* **Zero-shot geographic transfer.** Per-asset batch-perplexity distributions for TradeFM (trained exclusively on US equities) evaluated on held-out Jan 2025 data. Chinese (CN) median tracks US closely; Japanese (JP) median is $\sim2\times$ US, with substantial distributional overlap.

**Temporal Transfer.** A smaller TFM-ITCH variant trained on public NASDAQ ITCH 2020 data (Appendix D) achieves best-in-class inter-arrival fidelity on a 5-year forward test (2020 train $\rightarrow$ 2025 evaluation), spanning COVID-era volatility through post-inflation 2025 microstructure.

### 3.2. Stylized-Fact Reproduction

We test whether TradeFM-generated rollouts reproduce canonical stylized facts of financial returns. On log-return marginals, TradeFM achieves 2.3–3.1$\times$ lower KS distance than Compound Hawkes across return intervals 10–120s (Table 5, Appendix C); the ZI baseline is distributionally far from real data. Beyond marginals, TradeFM-generated rollouts reproduce canonical stylized facts (Cont, 2001): heavy-tailed returns (leptokurtic at short $\Delta t_r$), insignificant return autocorrelation, slow decay of absolute-return autocorrelation (volatility clustering), and aggregational Gaussianity (kurtosis approaches normal as $\Delta t_r$ grows); the ZI baseline fails on the first three (panels in Figure 5, numerical values in Table 5).

### 3.3. Order-Flow Distributional Fidelity

We measure distributional distance between TradeFM-generated and real order-flow features. On the broader order-flow distance suite (Table 2), TradeFM is the lowest-distance model on inter-arrival times, order-price depth, order-book imbalance, and ask volume. Two losses are concrete: *bid–ask spreads* (KS 0.238 vs Hawkes' 0.218) and *bid volume* (KS 0.386 vs Hawkes' 0.296). Hawkes processes explicitly model the inter-arrival dynamics that govern spread formation; TradeFM's spreads emerge indirectly from the orders the model places near the best bid and ask – the simulator only enforces price-time priority, so spread fidelity is a function of the model's tendency to generate quote-stuffing or liquidity-providing orders at the right depths. Finer volume binning (i.e. a larger output vocabulary) should aid performance here, at the cost of additional rare-token mass.

|  | KS distance | | | $W_1$ distance | | |
|---|---|---|---|---|---|---|
|  | ZI | Hawkes | TFM | ZI | Hawkes | TFM |
| Spread | 0.400 | **0.218** | 0.238 | 0.375 | **0.302** | 0.400 |
| IA time | 0.651 | 0.515 | **0.281** | 0.415 | 0.626 | **0.318** |
| Order depth | 0.436 | 0.281 | **0.169** | 0.390 | 0.348 | **0.339** |
| Book imbal. | 0.237 | 0.155 | **0.142** | 0.200 | 0.165 | **0.099** |
| Bid volume | 0.460 | **0.296** | 0.386 | 0.616 | 0.278 | **0.130** |
| Ask volume | 0.391 | 0.380 | **0.360** | 0.638 | 0.198 | **0.160** |

*Table 2.* Mean KS and $W_1$ distances on order-flow features, averaged over 9 assets across 3 liquidity tiers and 9 held-out months (81 asset-month contexts, 10 closed-loop rollouts each, $n$=810 rollouts). **Bold** marks the lowest distance per row. Log-return marginals at four $\Delta t_r$ are reported in Appendix Table 5.

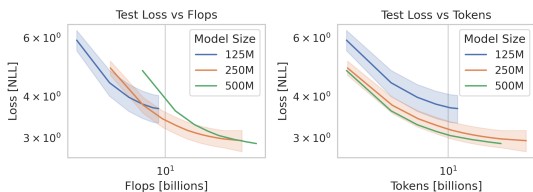

*Figure 3.* Test-loss scaling laws over the model family ($\alpha_C \approx 0.18$, $\alpha_D \approx 0.19$); full fit detail in Appendix C.

### 3.4. Additional Experiments

**Scaling.** Across the model family (125M, 250M, 500M; Figure 3), test loss follows clean power laws in both compute and data (Kaplan et al., 2020; Hoffmann et al., 2022). Our exponents ($\alpha_C \approx 0.18$, $\alpha_D \approx 0.19$) are 2–3$\times$ shallower than Chinchilla's language reference; we did not LR-tune by model size (all variants inherit $5\times10^{-5}$), so this gap may reflect domain structure or sub-optimally tuned training hyperparameters (e.g. $\mu$P-style transfer (Yang et al., 2022)).

**Conditioning-controllability validation** (liquidity tier, market-vs-participant) is in Appendix C.5.

## 4. Discussion

### 4.1. Limitations

We focus on US equities with zero-shot perplexity transfer to APAC; APAC stylized-fact rollouts, neural-baseline head-to-head comparison, downstream forecasting curves, and bootstrap confidence intervals on distributional metrics are deferred to subsequent rounds.

### 4.2. Future Directions

Downstream evaluation via open-loop forecasting and offline RL with TradeFM as a Markov decision process. Cross-asset-class extension to FX, fixed income, and crypto exploits the asset-agnostic representation (Section 2), with conditioning features (news, instrument metadata) added as input embeddings. GPU-accelerated simulators (Frey et al., 2023) unlock larger rollout sweeps.

**Disclaimer**

This paper was prepared for informational purposes by the Artificial Intelligence Research group of JPMorgan Chase & Co. and its affiliates ("JP Morgan") and is not a product of the Research Department of JP Morgan. JP Morgan makes no representation and warranty whatsoever and disclaims all liability, for the completeness, accuracy or reliability of the information contained herein. This document is not intended as investment research or investment advice, or a recommendation, offer or solicitation for the purchase or sale of any security, financial instrument, financial product or service, or to be used in any way for evaluating the merits of participating in any transaction, and shall not constitute a solicitation under any jurisdiction or to any person, if such solicitation under such jurisdiction or to such person would be unlawful.

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

# A. Background, Related Work, and Positioning

## A.1. Market Microstructure Primer

To provide context for a general AI/ML audience, we briefly introduce the core concepts of market microstructure fundamental to this work, which are standard features of modern electronic markets (Hasbrouck, 2007).

Financial markets are predominantly organized around a **Limit Order Book (LOB)**, a real-time record of all outstanding orders for a security that functions as a continuous, double-sided auction. It consists of a **bid** (buy) side and an **ask** (sell) side; the midpoint between highest bid and the lowest ask is an asset's **mid-price**. The ease with which an asset can be bought or sold quickly at a stable price is the asset's **liquidity**.

Market participants interact with the LOB through a sequence of actions, collectively known as **order flow**. Participants may submit **limit orders** with a specific price limit, which sit on the book waiting to be matched. The distance between the **order price** and the midprice is the **price depth**, quoted in **ticks** (the minimum price increment, typically $0.01) or **basis points** (0.01% of the price). They may also submit **market orders** for immediate execution against resting limit orders starting at the best bid/ask, and **cancellations** to withdraw resting orders. When an incoming order is matched with a resting one, a **fill** (or trade execution) occurs. This matching process is generally governed by a deterministic **price-time priority** algorithm, where orders are first prioritized by price and then by time of submission. These elements and mechanisms constitute **market microstructure**.

**Stylized facts as emergent properties.**  The strategic interactions of market participants give rise to endogenous market dynamics (Bouchaud, 2010). These dynamics, in turn, give rise to universal and persistent statistical properties known as **stylized facts**. These facts are observed across a wide range of assets, markets, and time periods, and serve as a crucial benchmark for the realism of any generative market model (Cont, 2001; Ratliff-Crain et al., 2025). Key stylized facts include:

- **Heavy-Tailed Returns**: returns are leptokurtic – extreme movements occur far more frequently than a Gaussian predicts.

- **Volatility Clustering**: high-volatility periods cluster together, manifesting as slowly decaying autocorrelation of absolute returns.

- **Lack of Autocorrelation in Returns**: consistent with efficient markets, return autocorrelation is insignificant beyond very short lags.

## A.2. Related Work on Market Simulation and LOB Models

The modeling of market microstructure has evolved from explicit, theory-driven formulations toward implicit, data-driven representation learning. Our work continues this trajectory, positioning a generative foundation model as the natural next step to learn universal market dynamics directly from raw, heterogeneous data.

**Classical stochastic models.**  A significant body of literature models order arrival times using point processes, such as Hawkes processes, to capture the self-exciting nature of order flow (Bacry et al., 2015). More sophisticated approaches adopt Compound Hawkes processes, which combine Hawkes-processes to model interarrival times with other fitted empirical distributions to model additional features like volumes and price depths (Jain et al., 2024). While providing strong theoretical grounding, these models rely on specific parametric assumptions (e.g., Gaussianity) that are unable to capture the heavy-tailed nature of market returns. Similarly, ensemble methods based on Hidden Markov Models have been applied to classify sequences of trade activity, demonstrating that probabilistic sequence models capture meaningful structure in financial data even with limited labeled examples (Kawawa-Beaudan et al., 2024).

**Agent-based models.**  Agent-based models simulate market dynamics by defining the behavior of individual participants and observing the emergent properties of the system (Byrd et al., 2020). While ABMs have historically required hand-crafting agent behaviors, recent approaches have shown success in calibrating agents on real market data (Dwarakanath et al., 2024). Our work contributes to this line of research by enabling the learning of complex market dynamics, which can serve as a foundation for more sophisticated agent-based modeling.

**Early deep learning models.** The application of deep learning to LOB data was pioneered by models like DeepLOB (Zhang et al., 2019). These models demonstrated the potential of learning features directly from data but were typically trained on a subset of instruments. This limits their ability to learn universal representations across diverse assets and market conditions.

**Transformers and foundation models in finance.** The Transformer architecture has been widely applied across domains including genomics (Ji et al., 2021), time-series forecasting (Wen et al., 2023), and payment transaction modeling (Raman et al., 2024). In market microstructure, recent transformer models focus on discriminative prediction for short-term forecasting (Berti & Kasneci, 2025; Xiao et al., 2025). These approaches operate on full limit order book snapshots and target single or few instruments.

A prominent generative approach is MaRS (Li et al., 2025), a market simulator with a foundation model backbone. While building on design principles from Li et al. (2025), TradeFM distinguishes itself in three dimensions (see body Table 1): (1) pre-training data explicitly constructed to **maximize heterogeneity** across thousands of assets, sectors, and liquidity regimes; (2) **partial observability** – learning from Level 3 trade messages rather than full LOB snapshots, matching the information available to real market participants; and (3) **zero-shot geographic transfer** (held-out perplexity) to unseen markets, enabled by scale-invariant feature design rooted in Universal Price Formation theory (Sirignano & Cont, 2021).

### A.3. TradeFM in the Structured-Data FM Landscape

**Tabular foundation models.** TabPFN (Hollmann et al., 2025) demonstrates in-context learning over tabular distributions but assumes IID rows with bounded categorical targets. TradeFM differs in three ways: rows are sequentially dependent (an event conditions on its predecessors), the joint vocabulary spans $\sim$16K mixed-radix tokens with a heavy-tailed empirical distribution, and inference is generative rather than predictive. The structural commonality – pre-train once, no per-instance calibration – carries over.

**Time-series foundation models.** Chronos (Ansari et al., 2024), TimesFM (Das et al., 2024), MOIRAI (Woo et al., 2024), and MOMENT (Goswami et al., 2024) discretize regularly-sampled univariate signals into bin tokens and pre-train on heterogeneous corpora for forecasting. TradeFM addresses the structurally distinct case of irregularly-sampled multi-feature event streams: each token is a five-feature joint draw rather than a single binned scalar, and time itself is a feature ($\Delta t_t$) rather than a sampling rate. The tokenization budget differs accordingly: Chronos uses $\sim$4,096 bins per scalar with temporal patching to amortize per-token compute; TradeFM uses 16,384 states per joint event with no temporal patching, trading higher per-event softmax cost for an event-aligned context that retains sub-millisecond timing.

**LOB-specific neural models.** DeepLOB (Zhang et al., 2019) and MaRS (Li et al., 2025) train on full limit-order-book snapshots, typically scoped to tens to hundreds of assets; LOBS5 (Nagy et al., 2023) trains on tokenized message flow with auxiliary L2 book-image conditioning at small scale (2 assets, 6.3M parameters); Kronos (Shi et al., 2025) aggregates to K-line OHLC bars rather than reconstructing the book. TradeFM diverges along two axes from each: *input scope* (raw L3 trade messages, no book-reconstruction or aggregation pipeline; mid-price estimated from execution prices, no privileged book-reconstruction access) and *coverage* (9,000+ assets, foundation-model scale). The trade-off is informational: snapshot-based models track standing-book state; TradeFM tracks only the events that modified the book. The latter sidesteps per-asset book-reconstruction overhead and is directly portable across markets with different snapshot conventions, evidenced by the bounded perplexity transfer to APAC markets (Appendix C).

**Benchmarks for structured-data FMs.** LOB-Bench (Nagy et al., 2025) packages distributional, discriminator, and impact pillars as a unified evaluation, but its discriminator-AUC pillar is reference-distribution dependent and its impact pillar requires intervention APIs that TradeFM's deterministic simulator supports natively. A full LOB-Bench run requires a snapshot-shaped output adapter we do not currently provide; we report KS and Wasserstein distances on event marginals as a strict subset of LOB-Bench's pillar (i). Closing this gap is a clear next step.

# B. Method Details

## B.1. Problem Formulation

We formulate the task of modeling market microstructure as a generative, autoregressive sequence modeling problem. Let the market dynamics be represented by a sequence of discrete trade events, $E = (e_1, e_2, \ldots, e_T)$. The objective is to learn the conditional probability distribution $P(e_t | e_{<t})$, where $e_{<t}$ denotes the sequence of events preceding $e_t$. By learning this distribution, the model can generate realistic sequences of future trade events, simulating the market's evolution.

**Trade event representation.** A single trade event $e_t$ is a multi-feature tuple capturing the state of the market at the moment of a transaction. Formally, an event is represented as $e_t = (\Delta t_t, \delta p_t, v_t, a_t, s_t)$, where the core features are: $\Delta t_t$: interarrival time since the previous event (seconds); $\delta p_t$: price depth of the transaction (basis points); $v_t$: volume of the transaction (shares); $a_t$: the action/order type (e.g., limit, cancellation); $s_t$: the side of the initiating order (buy/bid or sell/ask).

**Key technical challenges.** Modeling this data stream presents challenges inherent to high-frequency markets: the **Heterogeneity and Distribution Shift** across diverse assets and varying time periods; the **Sparsity and Irregularity** of the asynchronous event stream; the **Partial Observability** of the true market state from transaction data; and a **High-Dimensional, Multi-Modal Feature Space** combining continuous and categorical values.

## B.2. Data Sources and Scale

The model is pre-trained on a proprietary dataset built from billions of raw, tick-level US equities transactions, spanning 368 trading days from February 2024 to September 2025, across the breadth of the US equities market. This represents over 19 billion tokens across 1.9 million date-asset pairs. We employ a temporal hold-out strategy, reserving January 2025 onward across all assets for the test set, yielding a training set of 10.7 billion tokens and a test set of 8.7 billion tokens. The tokenizer is calibrated on the first 30 days of the training data, February 2024. For evaluating out-of-distribution generalization we also hold out one month of data from APAC regions, namely Jan. 2025, for both Japan and China.

## B.3. Mid-Price Estimation (EW-VWAP)

A robust estimate of the true market mid-price ($p_t^{\text{mid}}$) is critical for normalizing price-related features. While dedicated market data sources for this exist, they are often expensive. Given our access to raw transaction data, we seek to estimate this value directly. In our partial-information setting, we primarily observe the execution prices ($p_t^{\text{exec}}$) for consummated trades. The raw stream of $p_t^{\text{exec}}$ is a noisy version of the true mid-price $p_t^{\text{mid}}$.

A naive approach, such as a simple rolling average of execution prices, is insufficient. A fixed-width window of trades is not comparable across assets with different liquidity levels; a 50-trade window may span less than a second for a highly liquid asset but several hours for an illiquid one. A time-based window (e.g., 2 seconds) is more relevant, but a simple average still fails to account for trade volume. For example, an average that gives equal weight to a 1,000-share trade at $10.00 and a 1-share trade at $9.00 would produce a misleading estimate.

The conventional solution is the volume-weighted average price (VWAP), which is

$$\hat{p}_t^{\text{VWAP}} = \frac{\sum_{i=0}^{W} v_{t-i} p_{t-i}^{\text{exec}}}{\sum_{i=0}^{W} v_{t-i}} \tag{1}$$

To make this estimator more reactive to recent information, we introduce **Exponentially-Weighted Volume-Weighted Average Price (EW-VWAP)**. This is calculated by maintaining two separate exponential moving averages (EMAs): one for the volume-weighted price and one for the volume itself. For each incoming trade with execution price $e_t$ and volume $v_t$, we update the EMAs for the numerator ($N_t$) and denominator ($D_t$) as follows:

$$N_t = \alpha \cdot (p_t^{\text{exec}} \cdot v_t) + (1 - \alpha) \cdot N_{t-1}$$
$$D_t = \alpha \cdot v_t + (1 - \alpha) \cdot D_{t-1}$$

The EW-VWAP at time $t$ is then the ratio of these two values:

$$\hat{p}_t^{\text{EW-VWAP}} = \frac{N_t}{D_t} \tag{2}$$

The smoothing factor $\alpha$ is determined by a time-based halflife, ensuring that the estimate gives more weight to larger and more recent trades in a temporally consistent manner. This provides a stable and representative price benchmark that reflects the price at which the bulk of recent market activity has occurred. This estimator is used throughout the paper to normalize all price-related features to a common scale.

### B.4. Scale-Invariant Feature Construction

The statistical properties of trading data vary widely across sectors, liquidity profiles, and nominal prices. In raw dollar, share, and second terms, price depths, volumes, and interarrival times for an asset like AAPL may differ greatly from those of a penny stock. Trade representations must therefore be carefully designed to enforce homogeneity across assets. Sirignano & Cont (2021) demonstrate that price formation follows universal principles across stocks: universal models trained on all assets outperform asset-specific models, even for held-out stocks, suggesting deep learning can learn appropriate normalizations from raw data. Building on this, we explicitly construct scale-invariant features to enforce homogeneity, extending universality from order book contexts to heterogeneous event streams across diverse trading venues and market structures.

- **Interarrival Time** ($\Delta t_t$): wall clock time since the previous event: $w_t - w_{t-1}$, in seconds.

- **Log-Transformed Volume** ($v_t$): to compress the wide dynamic range of order sizes, which follow heavy-tailed, power-law distributions (Vyetrenko et al., 2020), we apply a logarithmic transformation: $v_t = \log(1 + V_t)$ where $V_t$ is the raw share volume.

- **Normalized Price Depth** ($d_t$): the depth of a limit order with order price $p_t^{\text{order}}$, relative to the mid-price: $d_t = (p_t^{\text{order}} - \hat{p}_t^{\text{mid}})/\hat{p}_t^{\text{mid}}$. This representation is comparable across differently priced assets, unlike prior work using price depths in ticks.

- **Relative Price Level vs. Open** ($\Delta p_t$): to capture intraday market movement, we quote the current mid-price relative to the day's opening price ($p_0$): $\Delta p_t = (p_t^{\text{mid}} - p_0)/p_0$.

While price-related features are computed as unit-less ratios, we refer to them in terms of basis points (bps) for interpretability, where a ratio of 0.01 corresponds to 100 bps.

### B.5. Mixed-Radix Tokenization: Algorithm

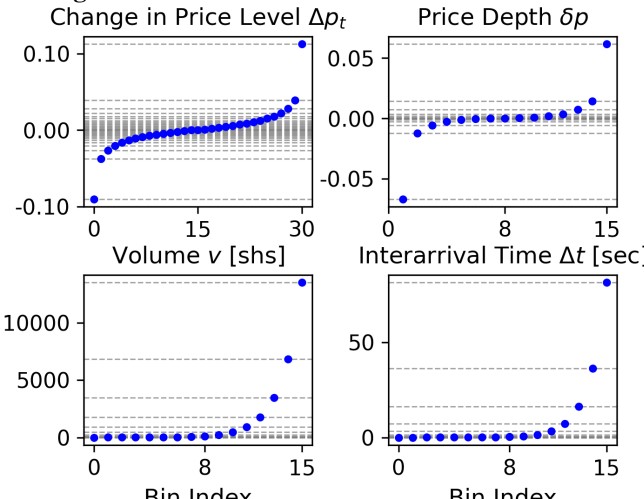

*Figure 4.* **Calibrated bin edges.** Price features (top) use quantile-based binning for high resolution near the mean; volume and time (bottom) use logarithmic bins to capture their wide dynamic range.

---

**Algorithm 1** Mixed-radix tokenizer: calibration on the first 30 days of training data, then per-event encoding.

---

**Input:** Trade events with features $\{\Delta t, \delta p, v, a, s\}$
**Input:** Bin counts $n_{\Delta t} = n_{\delta p} = n_v = 16$, $n_a = n_s = 2$
  1: **Calibration phase (once per training corpus):**
  2: **for** each continuous feature $f \in \{\Delta t, \delta p, v\}$ **do**
  3:     Remove NaN and infinite values from $f$
  4:     Compute upper-tail threshold $u = \text{percentile}(f, 99)$
  5:     **if** $f$ has signed support **then**
  6:         Compute lower-tail threshold $l = \text{percentile}(f, 1)$
  7:         Clip values outside $[l, u]$ to outlier bins
  8:     **else**
  9:         Clip values above $u$ to upper outlier bin
10:     **end if**
11:     Compute equal-frequency (quantile) bin edges $B_f$ within the clipped support
12: **end for**
13: **Encoding phase (per event):**
14: **for** each event $e$ **do**
15:     $I_{\Delta t}, I_{\delta p}, I_v \leftarrow \text{digitize}(e.\Delta t, e.\delta p, e.v)$ using $B_{\Delta t}, B_{\delta p}, B_v$
16:     $I_a, I_s \leftarrow$ binary indices for action and side
17:     $T_e \leftarrow I_a \cdot (n_s n_{\delta p} n_v n_{\Delta t}) + I_s \cdot (n_{\delta p} n_v n_{\Delta t}) + I_{\delta p} \cdot (n_v n_{\Delta t}) + I_v \cdot n_{\Delta t} + I_{\Delta t}$
18: **end for**
19: **return** Token sequence $\{T_e\}$ over vocabulary $|\mathcal{V}| = 16{,}384$

---

### B.6. Multi-Feature Token Composition

While our model's input at each time step is multi-featured, the decoder is trained to predict a single, unidimensional token, representing the core trade event. Thus, we combine the discrete bin indices of the trade-related features, $(i_{\Delta t}, i_{\delta p}, i_v, i_a, i_s)$, into a single composite integer, $i_{\text{trade}}$. This is accomplished by treating the feature indices as digits in a mixed base number system: each feature's bin index is a "digit", and the number of possible values for the subsequent features acts as the "base" at each position. With $n_{\delta p} = 16$ for price depth, $n_v = 16$ for volume, $n_{\Delta t} = 16$ for interarrival time, $n_s = 2$ (buy, sell) for side, and $n_a = 2$ (add or cancel order) for action type, the composite trade token is:

$$\begin{aligned} i_{\text{trade}} = & (i_a \times n_s \times n_{\delta p} \times n_v \times n_{\Delta t}) + \\ & (i_s \times n_{\delta p} \times n_v \times n_{\Delta t}) + \\ & (i_{\delta p} \times n_v \times n_{\Delta t}) + (i_v \times n_{\Delta t}) + i_{\Delta t} \end{aligned} \tag{3}$$

This yields a vocabulary size of $|\mathcal{V}| = 16{,}384$ for the predictable trade tokens. The model input at each time step is a tuple containing this token along with several non-predicted contextual features used for conditioning. These contextual features are provided as separate inputs (Appendix B.8), and are not part of the trade token $i_{\text{trade}}$ calculated in Equation (3):

- $i_l$: the liquidity bin index ($n_l = 3$), determined by binning each asset into low, medium, or high liquidity ranges based on its Average Daily Volume (ADV).

- $i_{\Delta p_t}$: the price-level-change bin index ($n_{\Delta p} = 32$).

- $I_{MP}$: a binary indicator distinguishing market-level sequences (order flow of the entire market) from participant-level sequences (order flow of individual participants); the training corpus contains both at an approximately 1.6:1 market-to-participant token ratio.

The final input is $[i_l, I_{MP}, i_{\Delta p_t}, i_{\text{trade}}]$. This formulation allows the model to be conditioned on broader market context while focusing its predictive power on the next trade event.

### B.7. Tokenization Worked Example

Given the imaginary sequence of trade events $e_t$ in Table 3, our features for timestep $t = 43$ are:

| Time-step | Time-stamp | Asset | Avg. Daily Vol. (shs) | Midprice ($) | Action | Side | Order Price ($) | Vol. (shs) |
|---|---|---|---|---|---|---|---|---|
| | | | | ⋮ | | | | |
| 42 | 09:45:30 | AAPL | 53,496,022 | 182.45 | ADD | BUY | 182.44 | 500 |
| 43 | 09:45:38 | AAPL | 53,496,022 | 182.48 | ADD | SELL | 182.50 | 750 |
| 44 | 09:45:52 | AAPL | 53,496,022 | 182.50 | CANCEL | BUY | 182.49 | 300 |
| | | | | ⋮ | | | | |

*Table 3.* **Example trade sequence.** Toy example of trading activity for an imaginary AAPL sequence, demonstrating the multi-feature and heterogeneous nature of our data pre-tokenization.

- $\Delta t_t = w_t - w_{t-1} = 09{:}45{:}38 - 09{:}45{:}30 = 8$ sec

- $\delta p_t = (p_t^{\text{order}} - p_t^{\text{mid}})/p_t^{\text{mid}} = (\$182.50 - \$182.48)/\$182.48 = 0.011\% = +1.1$ bps

- $v_t = 750$ shs

- $a_t =$ Add Order

- $s_t =$ Sell

Using our calibrated bins, we discretize to $i_{\Delta t_t} = 11$, $i_{\delta p_t} = 7$, $i_{v_t} = 7$, $i_{a_t} = 0$, $i_{s_t} = 1$. Applying the mixed-radix encoding gives composite token $i_{\text{trade}} = 6{,}011$. Assuming an opening price $p_0 = \$179.50$, the price-level-change feature is $\Delta p_t = 0.017\% = +1.7$ bps, discretizing to $i_{\Delta p_t} = 19$. The asset's average daily volume of 53M places it in the high-liquidity bin ($i_l = 2$). Treating this as a market-level sequence ($I_{MP} = 0$), the final model input is $[i_l, I_{MP}, i_{\Delta p_t}, i_{\text{trade}}] = [2, 0, 19, 6011]$.

### B.8. Architecture and Training Configuration

**Tabular Input Embedding.** We employ a tabular embedding approach to handle our multi-feature input tokens (as described in Appendix B.6). Each feature in the input tuple $[i_l, I_{MP}, i_{\Delta p_t}, i_{\text{trade}}]$ is first projected into its own embedding space using an embedding table. These embedding vectors are concatenated and passed through a linear projection layer to create a unified representation in the model's hidden dimension.

TradeFM is a decoder-only Transformer, trained from scratch with a custom configuration. The architecture is based on the Llama family (Touvron et al., 2023) and incorporates modern enhancements for efficiency and performance, including:

- **Grouped-Query Attention (GQA):** balances the performance of Multi-Head Attention with the reduced memory bandwidth of Multi-Query Attention, enabling faster inference and larger batch sizes.

- **Rotary Position Embeddings (RoPE):** encodes relative positional information by applying a rotation to query and key vectors, which has been shown to improve generalization for long sequences.

**Hyperparameters and scaling.** The model size is guided by the Chinchilla scaling laws, which suggest a compute-optimal ratio of approximately 20 training tokens per model parameter (Kaplan et al., 2020; Hoffmann et al., 2022). Given our dataset of 10.7 billion tokens, this implies a target model size of around 525 million parameters. Our final hyperparameters are:

- Context Length: 1,024 tokens

- Hidden Layers: 32

- Embedding Dimension: 1,024

- Intermediate MLP Size: 4,096

- Attention Heads: 32

- Key-Value Heads (GQA): 8 heads, 4 groups

- Total Parameters: 524.4 million

**Training configuration.** We train the model on an AWS instance with 3 Nvidia A100 GPUs, each with 80GB of RAM. All training is performed in `fp16` half-precision. To achieve an effective batch size of 4,032, we use a per-device batch size of 24 and gradient accumulation over 56 steps. For further memory optimization and training acceleration, we use the Accelerate library. The model is trained using the AdamW optimizer with a linear learning rate schedule, a learning rate of $5 \times 10^{-5}$, and 500 steps of warmup. Following recommendations for training on large datasets (Muennighoff et al., 2023), we train for a total of 4 epochs. Closed-loop rollouts at inference use multinomial sampling with a repetition penalty. Different model sizes use slightly different training configurations summarized in Table 4.

| Model Size | Num. Train Tokens | Batch Size | GPUs | Train Time / Epoch (hrs) | Optimization |
|---|---|---|---|---|---|
| 125M | 2.6B | 24 | 3×A100 | 17 | Accelerate |
| 250M | 6.4B | 32 | 4×A10G | 29 | DeepSpeed |
| 500M | 10.7B | 24 | 3×A100 | 128 | Accelerate |

*Table 4.* **Training configuration.** Setup details for different model sizes, including token counts, batch sizes, hardware, and training time per epoch.

### B.9. Closed-Loop Simulator Pseudocode

**Algorithm 2** Market Simulator: Part 1 – Initialization and Utilities

**Input:** Sequence of order transactions, initial price $p_0$, simulation parameters
 1: **Initialize Exchange:**
 2: Set initial price $p_0$
 3: Initialize order book, midprice, fills, deletes, spreads, bid/ask volumes
 4: **Function:** INITIALIZEORDERBOOK(order_columns)
 5: Reset order book, midprice, fills, deletes, spreads, bid/ask volumes
 6: Set initial bid/ask to $p_0$
 7: **Function:** GETORDERPRICE(transaction)
 8: **if** order is market **then**
 9:    **if** side is Sell **then**
10:       price ← lowest ask
11:    **else**
12:       price ← highest bid
13:    **end if**
14: **else**
15:    price ← (order price depth / 10,000) × current midprice + current midprice
16: **end if**
17: **Return** price
18: **Function:** GENERATEFILL(best_past_order, order, quantity)
19: Compute time since best_past_order
20: Determine match price based on order/best_past_order types (market vs limit)
21: **Return** fill record with IDs, sides, prices, depths, volume, time delta

---

**Algorithm 3** Market Simulator: Part 2 – Step Functions

---

**Function:** STEPORDERBOOK(order)
Extract side and price from order
**while** order volume $> 0$ **do**
    Find matching opposite-side orders by price-time priority
    **if** no matching orders **then**
        **break**
    **end if**
    Select best matching order (highest bid or lowest ask)
    **if** best_past_order volume $>$ order volume **then**
        Reduce best_past_order volume by order volume; record fill; **return**
    **else if** best_past_order volume $<$ order volume **then**
        Reduce order volume by best_past_order volume; remove from book; record fill
    **else**
        Record fill; remove best_past_order from book; **return**
    **end if**
**end while**
**if** order volume $> 0$ **then**
    Add partially filled order to book
**end if**
**Function:** STEPMIDPRICE(transaction)
Update best bid/ask from book; midprice $\leftarrow$ average of best bid/ask
**Function:** STEPSIM(transaction)
**if** action is Add **then**
    Compute order price; update midprice; step order book
**else if** action is Delete **then**
    Match on order ID; remove matching orders; record deletes; update midprice
**end if**
**Function:** RUNSIMULATION(data)
Initialize order book
**for** each transaction in data **do**
    StepSim(transaction)
**end for**
**return** fills and midprice history

---

### B.10. Baselines: Zero-Intelligence and Compound Hawkes

**Zero-Intelligence (ZI) agent.** The Zero-Intelligence agent is a canonical null model used to test whether a model learns complex, conditional dynamics beyond the market's basic structural properties (Gode & Sunder, 1993; Farmer et al., 2005). To provide a fair baseline, our ZI agent generates orders stochastically by sampling from distributions calibrated to match the marginals of key features in a 450-million-trade sample of the training data. Specifically, side and action type are sampled from their empirical categorical distributions; interarrival time and order volume are each sampled from a fitted Exponential; and price depth is drawn from a Gaussian Mixture Model (GMM). The resulting ZI agent orders are processed through the identical market simulator and evaluation pipeline as TradeFM to ensure a direct and fair comparison. We compute 2,048 rollouts of 1,024 events, and compute the same stylized facts.

**Compound Hawkes baseline.** Hawkes Processes are commonly applied to market data for their ability to robustly model interarrival times of self-exciting events (Bacry et al., 2015; Jain et al., 2024). We adopt the Compound Hawkes model which combines a Hawkes process for modeling interarrival times with empirical distributions for modeling additional event features like volume and price depth. We use the same 450M-trade data as is used to train our zero-intelligence baseline, and separate the data based on action and side. We then fit a Hawkes process using a sum-of-exponentials kernel, with 4 dimensions, one for each combination of action and side (buy-delete, buy-add, sell-delete, sell-add). For each of these action-side combinations we calibrate a Gaussian Mixture Model for price depths, and an Exponential for volume.

# C. Extended Experimental Results

### C.1. Log-Return Distributional Fidelity

| $\Delta t_r$ (s) | KS distance | | | $W_1$ distance | | |
|---|---|---|---|---|---|---|
| | ZI | Hawkes | TFM | ZI | Hawkes | TFM |
| 10 | 0.376 | 0.039 | **0.013** | 0.0006 | 0.0004 | **0.0001** |
| 30 | 0.439 | 0.075 | **0.024** | 0.0015 | 0.0012 | **0.0002** |
| 60 | 0.429 | 0.101 | **0.035** | 0.0027 | 0.0023 | **0.0004** |
| 120 | 0.429 | 0.098 | **0.043** | 0.0043 | 0.0043 | **0.0007** |

*Table 5.* Mean KS and $W_1$ distances on log-return marginals at four return intervals $\Delta t_r$, averaged over 9 assets $\times$ 3 liquidity tiers $\times$ 9 months. KS ratio Hawkes / TradeFM at the four intervals: 3.0, 3.1, 2.9, 2.3 – giving the 2.3–3.1$\times$ envelope referenced in the body.

### C.2. Stylized Facts of Generated Returns

Our simulations reproduce four canonical stylized facts of log returns (Cont, 2001):

- **Lack of autocorrelation**: the autocorrelation of simulated log returns quickly decays to statistically insignificant levels as the lag $\tau$ increases. The ZI baseline exhibits spurious autocorrelation.

- **Long-range dependence**: the autocorrelation of absolute log returns decays slowly, indicating that our model has captured the long-memory nature of volatility clustering.

- **Heavy tails**: the kurtosis of simulated returns is high for short time scales ($\Delta t_r$), confirming the presence of heavy tails. TradeFM most faithfully captures the leptokurtic nature of returns and its decay across time scales.

- **Aggregational Gaussianity**: as $\Delta t_r$ increases, the kurtosis of TradeFM correctly approaches that of a normal distribution, capturing the reversion towards normality over longer time horizons.

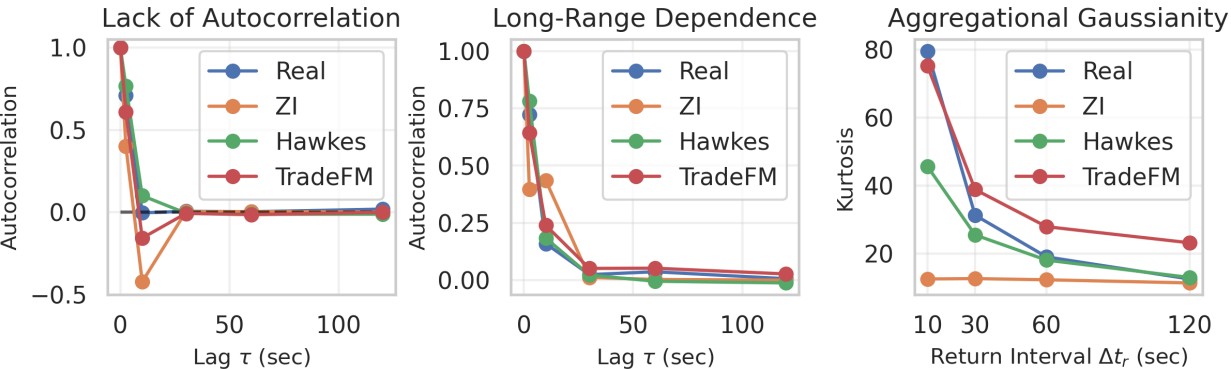

*Figure 5.* **TradeFM stylized-fact validation.** Simulated returns exhibit: (left) near-zero autocorrelation, (middle) slowly decaying autocorrelation of absolute returns (volatility clustering), and (right) heavy tails and aggregational Gaussianity.

### C.3. Geographic Out-of-Distribution Evaluation

We evaluate the model, trained exclusively on US equities, on a hold-out set of assets from APAC markets (China and Japan). We detail these held-out datasets in Table 6.

| Country | Assets | Date-Asset Pairs | Tokens |
|---|---|---|---|
| US (held-out) | 6,885 | 81,203 | 857,017,219 |
| China | 4,926 | 68,925 | 37,408,529 |
| Japan | 2,932 | 37,235 | 286,476,052 |

*Table 6.* **Geographic held-out datasets.** Dataset statistics for US, China, and Japan. All geographies are evaluated on Jan. 2025 data.

We report per-asset *batch perplexity*: the model is fed the first half of each batch as context and computes perplexity on the second half. Median per-asset batch perplexity is $\sim$17 (US), $\sim$18 (CN), $\sim$34 (JP); 75th-percentile is $\sim$25 (US), $\sim$27 (CN), $\sim$46 (JP); see Figure 2 (body) for the full distributions. We caution that perplexity transfer is a marginal-distribution check; stylized-fact reproduction (volatility-clustering shape, return autocorrelation, kurtosis vs $\Delta t_r$) on APAC markets is the more falsifiable transfer test and is not measured here.

### C.4. Temporal Drift and Stationarity

As financial markets are dynamic and market regimes are constantly changing, we investigate the tendency of model performance to drift over time. Our tokenizer's main contribution is to standardize representations of market features over both the liquidity and time regime. Relative (scale-invariant) construction is the design choice under test: Figure 6 contrasts tick-based and relative constructions of the sample feature $\Delta p_t$ across liquidity profiles.

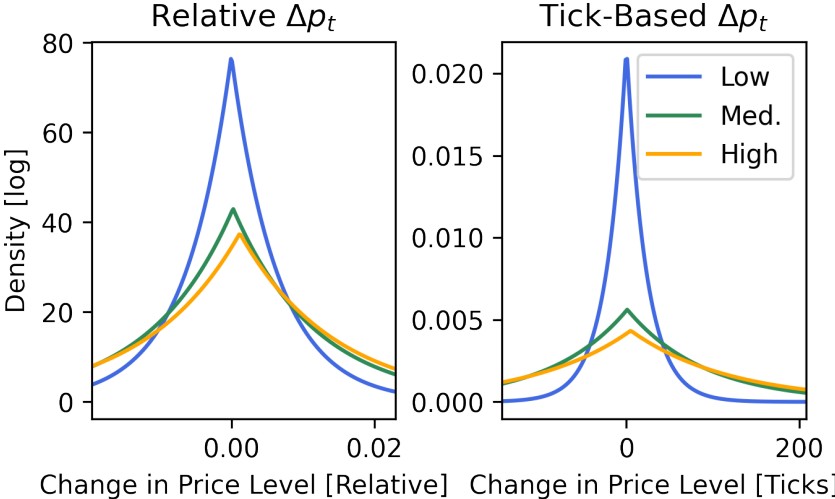

*Figure 6.* **Tick-based vs. relative features.** Properties of tick-based vs. relative feature construction for the sample feature $\Delta p_t$, across liquidity profiles. Relative features generalize better across assets than absolute, tick-based features.

In Figure 7 we demonstrate the universality of these features by exploring the distribution of our relative price level, relative price depth, interarrival time, and volume features in both the month used to calibrate our tokenizer (Feb. 2024) and one year later (Feb. 2025). We observe that our features are stationary over this period even as volatility, price level, and other market conditions vary. Figure 8 shows the Kolmogorov-Smirnov and Wasserstein distance of each of these features between the tokenizer calibration month and each of 9 held-out months, including a non-stationary feature (raw mid-price) for context.

### C.5. Controllability Experiments

We verify that the model respects its conditioning tokens and that its output can be reliably steered via the indicator features ($I_{MP}$ and $i_l$). We generate 256 context-free trajectories of 512 tokens each, for every combination of market-participant and liquidity indicators. We then analyze the statistical properties of the raw generated orders by computing the standard deviation of volumes and interarrival times for each condition.

Figure 9 shows two trends:

- The variance of both volume and interarrival time is consistently higher for market-level generation than for participant-level, aligned with the intuition that the aggregate behavior of an entire market is inherently more volatile than the behavior of a single participant.

- The model captures linear relationships between liquidity and the variance of interarrival times and volumes.

Collectively, these results demonstrate that TradeFM has learned a generalizable, conditional model of market behavior, capable of generating statistically and contextually appropriate order flow.

### C.6. Scaling-Law Fits (Extended)

We trained models $\in \{125M, 250M, 500M\}$. The 500M parameter model is our largest and serves as the primary evaluation target throughout this paper. The scaling law plots in Figure 10 demonstrate the expected power-law relationships between compute, dataset size, and test loss. These plots include repeated data, as we train for four epochs. We compute the minimum loss frontier in terms of both compute and dataset size, and fit power laws to find that the test loss $L(C)$ with respect to

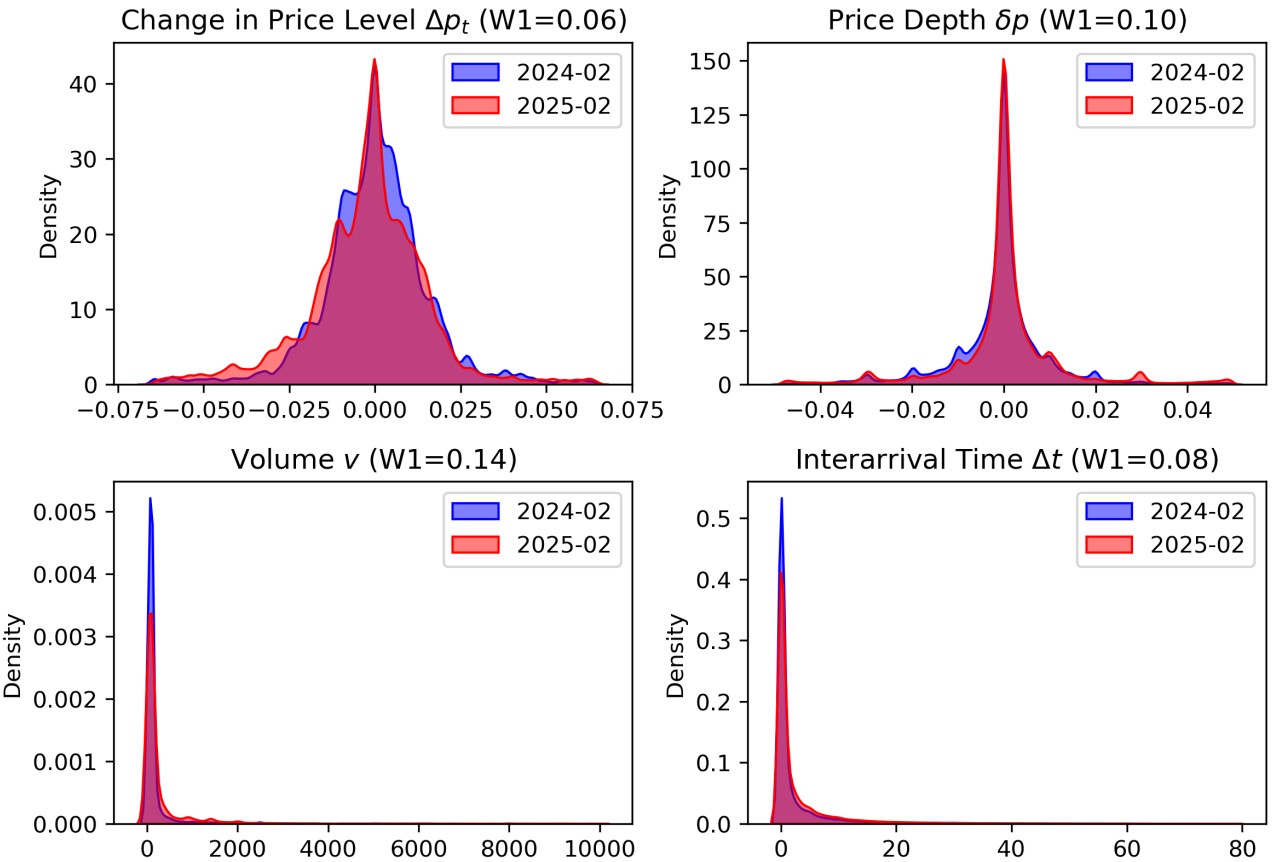

*Figure 7.* **Feature stationarity over time.** Kernel-density estimation of feature distributions from the tokenizer calibration period of Feb. 2024 to one year later in Feb. 2025. Our feature engineering successfully makes these features stationary, enabling generalization to out-of-distribution temporal regimes.

compute in FLOPs $C$, and $L(D)$ with respect to dataset size in tokens $D$, follow:

$$L(C) \propto C^{-\alpha_C}, \quad \alpha_C \approx 0.18$$

$$L(D) \propto D^{-\alpha_D}, \quad \alpha_D \approx 0.19$$

These exponents are 2–3× shallower than Chinchilla's language-model reference ($\alpha_C \approx 0.28$). We trained all three variants at the same peak learning rate ($5 \times 10^{-5}$) inherited from the 500M target; the 125M anchor is therefore likely under-tuned. A $\mu$P-style learning-rate transfer (Yang et al., 2022) or a learning-rate sweep at 125M would tighten the exponent estimate; we leave this as an open methodological question rather than an established domain claim.

While 500M is small relative to general purpose LLMs (Llama-3 8B, GPT-OSS 20B), it is large for the financial microstructure domain, and similar to other domain-specific models such as MaRS. Standard prior models in this field typically have <10M parameters (e.g., DeepLOB 60K, LOBS5 6.3M). TradeFM represents a >50× increase in model capacity over existing domain-specific baselines.

## D. TFM-ITCH: Public-Data Temporal Robustness

### D.1. Setup

TFM-ITCH is a separate 90M-parameter variant trained on the public NASDAQ ITCH 2020 corpus (1.8B tokens, 8,159 assets), using the identical Feb-2024 tokenizer calibration and the same proprietary 2025 test set as TFM-500M for an apples-to-apples comparison. The 5-year forward gap from 2020 training data to 2025 evaluation spans the COVID volatility regime through post-inflation 2025 microstructure.

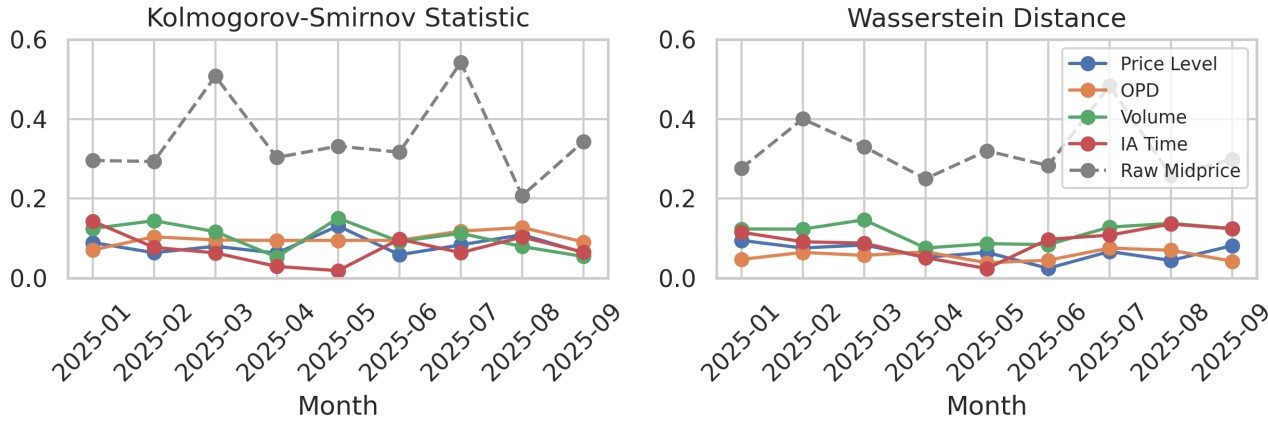

*Figure 8.* **Tokenizer drift over months.** Kolmogorov-Smirnov and Wasserstein distances between feature distributions during the tokenizer calibration month and held-out months. Raw mid-price, a non-stationary feature, is included for context.

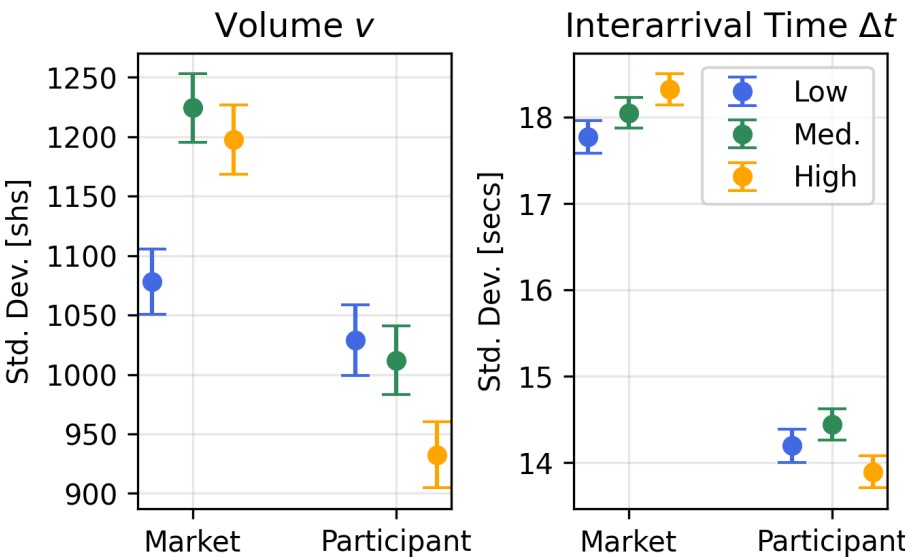

*Figure 9.* **Controllability experiments.** Standard deviation of generated volumes and interarrival times, conditioned on liquidity ($i_l$) and observation level ($I_{MP}$). The model produces statistically distinct order flow per condition, demonstrating controllable generation.

### D.2. Distributional Fidelity

| $\Delta t_r$ (s) | KS (TFM-ITCH) | $W_1$ (TFM-ITCH) |
|---|---|---|
| 10 | 0.066 | 0.0003 |
| 30 | 0.130 | 0.0009 |
| 60 | 0.190 | 0.0018 |
| 120 | 0.226 | 0.0031 |

*Table 7.* **ITCH log-return distributional fidelity.** TFM-ITCH evaluated on the same proprietary 2025 test set used for TFM-500M.

### D.3. Discussion

TFM-ITCH achieves best-in-class inter-arrival fidelity across the panel (KS $0.191 \pm 0.085$ vs TFM-250M $0.254$ and Hawkes $0.515$) despite a 5-year gap between training data (2020) and evaluation (2025), spanning multiple market regimes. Honest weaknesses: bid and ask volumes underperform TFM-500M, TFM-250M, and Hawkes on KS – likely a combination of TFM-ITCH's smaller parameter count (90M vs TFM-500M's 524M) and public ITCH-derived volume signals not fully calibrating the simulator's volume-arrival behavior at higher fidelity. The headline reading is that the same model behavior on inter-arrival dynamics emerges across an independent training corpus and a half-decade horizon – evidence that the mixed-radix scheme captures something genuine about event-stream structure, not artifacts of a single dataset or training

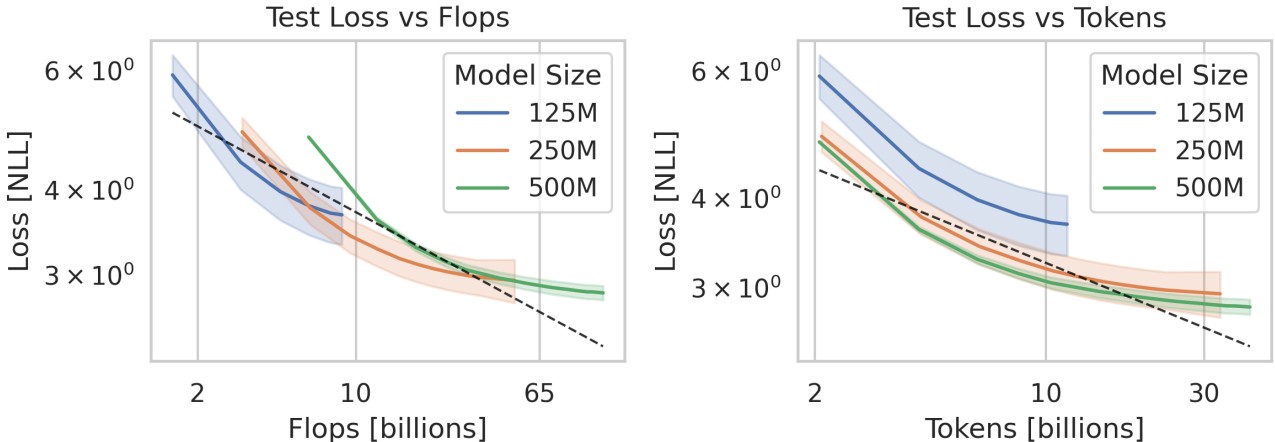

*Figure 10.* **Scaling law analysis.** Test loss (negative log likelihood) on held-out data one month in advance of the training data cutoff. The black dashed line represents the power law fit to the minimum loss frontier.

| Metric | TFM-500M | TFM-250M | TFM-ITCH | Hawkes | ZI |
|---|---|---|---|---|---|
| Spreads | 0.238 | $0.245 \pm 0.077$ | $0.335 \pm 0.084$ | $\mathbf{0.218 \pm 0.049}$ | $0.400 \pm 0.073$ |
| IA Times | 0.281 | $0.254 \pm 0.115$ | $\mathbf{0.191 \pm 0.085}$ | $0.515 \pm 0.104$ | $0.651 \pm 0.107$ |
| Price Depths | **0.169** | $0.170 \pm 0.057$ | $0.175 \pm 0.046$ | $0.281 \pm 0.074$ | $0.436 \pm 0.068$ |
| OBI | **0.142** | $0.120 \pm 0.038$ | $0.197 \pm 0.068$ | $0.155 \pm 0.050$ | $0.237 \pm 0.066$ |
| Bid Volume | 0.386 | $0.340 \pm 0.061$ | $0.593 \pm 0.074$ | $\mathbf{0.296 \pm 0.091}$ | $0.460 \pm 0.106$ |
| Ask Volume | 0.360 | $0.330 \pm 0.079$ | $0.571 \pm 0.065$ | $\mathbf{0.380 \pm 0.054}$ | $0.391 \pm 0.062$ |

*Table 8.* **Order-level distributional fidelity (KS, mean $\pm$ std over 9 months).** TFM-500M is the per-asset-month panel mean from Table 2 (per-month std not reported). TFM-ITCH achieves best-in-class IA times, ties on price depths; honest losses on bid/ask volumes vs TFM-500M / TFM-250M / Hawkes.

regime.

## E. Preliminary Downstream Explorations

We close with preliminary explorations of the downstream applications the TradeFM–simulator system is built toward; systematic downstream evaluation remains future work.

The most immediate applications are forecasting and learning-based execution. Closed-loop rollouts directly yield plausible, multi-step forecasts of future market trajectories (Figure 11), and a task-specific forecaster can be fine-tuned from the pre-trained model. For optimal execution, RL agents can be post-trained against the world model, learning to execute large orders while minimizing costs such as price impact and the bid-ask spread.

The integrated system also serves as an environment for "what-if" analysis and stress testing – studying systemic risk and market stability under controlled conditions. Regulators and risk managers (Dwarakanath et al., 2024) can simulate the market's response to extreme or counterfactual scenarios by injecting large, anomalous orders into a historical context and observing the resulting price trajectory. Figure 12 demonstrates this: for a sample asset, we inject buy or sell orders at $10\times$ the frequency found in the real context and average mid-price forecasts over 10 rollouts. Injecting sell orders drives the mid-price down; injecting buy orders drives it up.

## F. Reproducibility

Architecture, hyperparameters, training configuration, tokenizer pseudocode (Algorithm 1), and simulator pseudocode (Algorithms 2 and 3) are documented in Appendix B. The proprietary US-equities corpus used for TFM-500M / TFM-250M training is not publicly redistributable; the TFM-ITCH variant (Appendix D) is trained on NASDAQ ITCH data, which is publicly available but requires acquisition through NASDAQ. KS and 1-Wasserstein distances are reported as means over the 9 assets (across 3 liquidity tiers) $\times$ 9 held-out months $\times$ 10 rollouts panel.

| Metric | TFM-500M | TFM-250M | TFM-ITCH | Hawkes | ZI |
|---|---|---|---|---|---|
| Spreads | 0.400 | 0.391 ± 0.190 | 0.561 ± 0.175 | **0.302 ± 0.107** | 0.375 ± 0.191 |
| IA Times | **0.318** | 0.417 ± 0.191 | 0.401 ± 0.198 | 0.626 ± 0.157 | 0.415 ± 0.119 |
| Price Depths | 0.339 | **0.313 ± 0.066** | 0.321 ± 0.103 | 0.348 ± 0.079 | 0.390 ± 0.106 |
| OBI | **0.099** | 0.099 ± 0.046 | 0.139 ± 0.076 | 0.165 ± 0.045 | 0.200 ± 0.056 |
| Bid Volume | **0.130** | 0.187 ± 0.052 | 0.202 ± 0.108 | 0.278 ± 0.094 | 0.616 ± 0.080 |
| Ask Volume | **0.160** | 0.194 ± 0.094 | 0.155 ± 0.051 | 0.198 ± 0.072 | 0.638 ± 0.114 |

*Table 9.* **Order-level distributional fidelity ($W_1$, mean ± std over 9 months).** TFM-500M is the per-asset-month panel mean from Table 2 (per-month std not reported).

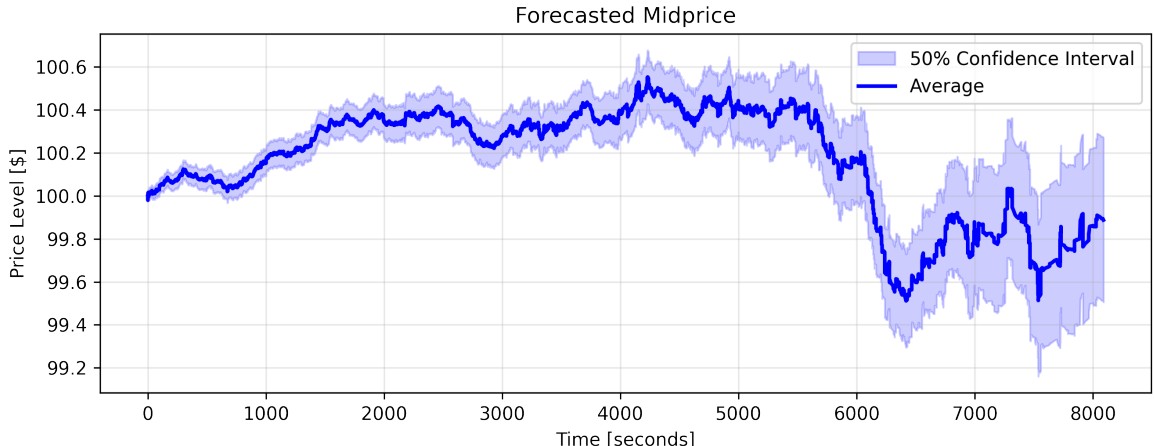

*Figure 11.* **Multi-step mid-price forecast.** Rollout-based forecast for an imaginary asset initially priced at $100. The average trajectory and 50% confidence interval over 256 simulations show the model generates plausible, non-stationary market paths.

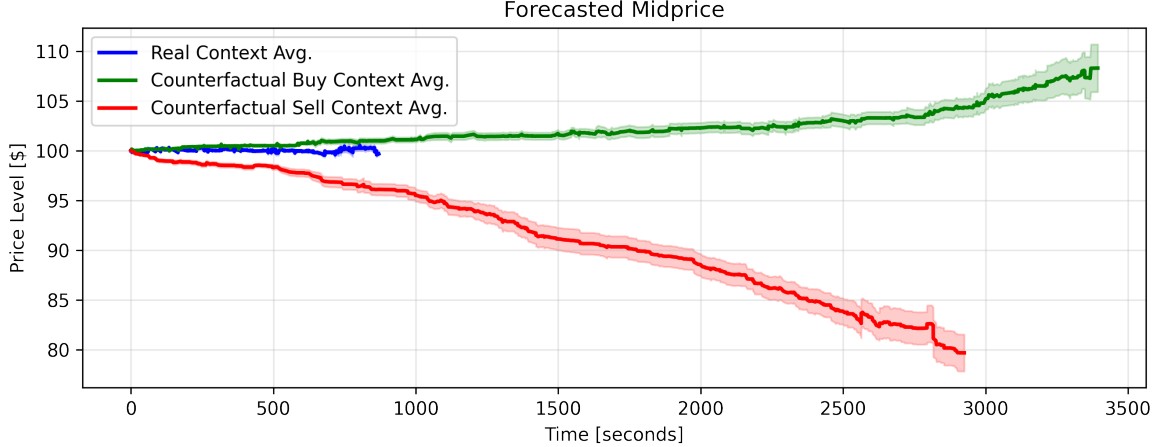

*Figure 12.* **Counterfactual stress testing.** The model's generated price path responds realistically to injected anomalous order flow ($10\times$ normal frequency), demonstrating its utility for impact analysis.

