# OpenReview forum: "TradeFM: A Generative Foundation Model for Trade-flow and Market Microstructure"
_ICML.cc/2026/Workshop/FMSD — FMSD @ ICML 2026 Poster_

### Official Review · Reviewer_tLPp · 2026-05-19
**A promising large-scale trade foundation model**

**Rating:** 7
**Confidence:** 3

**Review:**

Summary: The paper introduces TradeFM, a 524M-parameter decoder-only Transformer for generative modeling of market microstructure from Level-3 trade/order-flow messages. It proposes a scale-invariant representation of irregular, mixed-type trade events using inter-arrival time, normalized price depth, volume, action, and side, then combine discretized feature bins into a single mixed-radix token vocabulary. The model is trained on 10.7B tokens from more than 9,000 US equities and paired with a deterministic limit-order-book simulator to generate closed-loop market rollouts. Empirically, the paper evaluates held-out perplexity, geographic transfer to Chinese and Japanese markets, reproduction of financial stylized facts, order-flow distributional fidelity against zero-intelligence and Compound Hawkes baselines, scaling-law behavior over three model sizes, and a smaller public-data ITCH variant for temporal robustness.

Strong Points:

1. This work is ambitious and clearly relevant to structured-data foundation modeling.

2. Training a 524M-parameter Transformer on billions of trade messages across more than 9,000 equities is a substantial engineering and modeling effort. The breadth of the training corpus is one of the paper's most compelling aspects, especially compared with existing works trained on a small number of instruments or reconstructed snapshots.

3. The representation based on normalized price depth, log volume, inter-arrival time, action, and side is a reasonable attempt to make heterogeneous assets comparable.

Potential Weaknesses:

1. The paper does not provide detailed ablation studies for key modeling choices.

2. The closed-loop evaluation compares mainly against zero-intelligence and Compound Hawkes processes. These are useful baselines, but the paper does not establish superiority over modern neural generators in the VAE/diffusion families.

---

### Official Review · Reviewer_f7e4 · 2026-05-19
**Promising and relevant, but empirically under-validated for its foundation-model claims**

**Rating:** 6
**Confidence:** 4

**Review:**

**Summary**

This paper introduces TradeFM, a 524M-parameter decoder-only Transformer for generative modeling of market microstructure from Level-3 message data. The paper proposes scale-invariant feature construction, a mixed-radix tokenization scheme that maps multi-feature order events into a single discrete token, and a closed-loop simulation framework in which generated events are executed by a deterministic limit-order-book simulator. The model is trained on a large proprietary corpus covering more than 9,000 US equities and is evaluated through perplexity, stylized-fact reproduction, and distributional distances such as KS and Wasserstein metrics.

Overall, the submission is highly relevant to the workshop theme and addresses an important structured-data foundation-model problem. However, I find the empirical validation too limited for the strength of the claims, especially given that the paper presents TradeFM as a foundation model for market microstructure. The current experiments do not yet convincingly demonstrate broad generalization, practical utility, or superiority over sufficiently strong baselines.




**Strengths**

The paper tackles an important and timely problem: building generative foundation models for irregular, mixed-type, event-level financial market data. This is a strong fit for the workshop, since market microstructure data differs substantially from standard tabular or regularly sampled time-series data.

The proposed representation is interesting. The scale-invariant feature design and mixed-radix tokenization provide a clean way to convert heterogeneous event streams into a Transformer-compatible discrete sequence.

The scale of the training corpus is impressive. Training on billions of events across thousands of US equities is a significant engineering contribution, and the attempt to evaluate zero-shot transfer to non-US markets is valuable.

The paper also goes beyond pure next-token loss by evaluating generated rollouts using stylized facts and distributional metrics.



**Areas for Improvement**

The main weakness is that the empirical evaluation is too narrow relative to the paper’s foundation-model claims. Although the model is trained across more than 9,000 assets, the main closed-loop realism evaluation appears to be conducted on only 9 assets across 3 liquidity tiers and 9 held-out months. The paper does not clearly specify which 9 assets were selected, how they were selected, whether they are representative, or whether the results are robust to alternative asset samples. This makes the evaluation difficult to interpret and hard to reproduce.

The out-of-distribution APAC evaluation is also limited. The paper reports zero-shot transfer mainly through perplexity distributions, but does not provide closed-loop rollouts, stylized-fact reproduction, forecasting performance, or downstream utility on those markets. Perplexity overlap is encouraging, but it is not enough to establish that the model generates realistic market dynamics in geographically different market structures.

The paper does not demonstrate practical usefulness. The claimed applications include synthetic data generation, stress testing, market impact analysis, and learning-based trading agents, but the experiments do not include forecasting curves, trading/profitability tests, market-impact evaluation. As a result, it remains unclear whether TradeFM is useful beyond matching selected marginal distributions.

There are also important clarity issues around the data semantics. The paper alternates between terms such as trade messages, order events, L3 messages, market-level sequences, and participant-level sequences. However, it is not always clear whether the data consists of market-wide L3 order messages, participant-level private order flow, trade executions, or a mixture of these. This distinction is important because the information available in a market-level L3 feed is very different from the information available to a single participant. The market-vs-participant indicator further raises questions: What exactly is a participant-level sequence? Who is the participant? Is it a single institution, multiple anonymized participants, or a constructed subset? What proportion of training data is market-level versus participant-level?

The simulator mechanics also need clarification, especially for cancellations. The generated token appears to include time, price depth, volume, action, and side, but not an order identifier. However, realistic cancellation requires identifying which resting order is being cancelled. If cancellations are matched heuristically by price, side, volume, or time priority, this should be explicitly described and evaluated. If order IDs are used internally, the relationship between order IDs and the generated token representation must be clarified.

The baseline comparison is not strong enough. Zero-intelligence and Compound Hawkes models are useful classical anchors, but they are not sufficient for a paper making foundation-model claims. The paper should compare against stronger statistical and neural baselines, or at least include adapted versions of existing LOB/message-flow generative models in the same output space.

Finally, several important ablations are missing. Since scale-invariant representation and mixed-radix tokenization are central contributions, the paper should empirically show that these choices matter. Without such ablations, it is difficult to know whether the reported performance comes from the proposed representation, the model scale, the simulator, or the limited evaluation setup.

**Detailed Comments**


The paper trains on a very broad asset universe but evaluates closed-loop realism on only 9 assets. Please specify the selected assets, the selection procedure, their liquidity tiers, sectors, price ranges, and volatility regimes. I strongly recommend expanding the evaluation to a much larger and more diverse set of assets, or at least reporting confidence intervals over multiple randomly sampled asset panels. This is important to rule out cherry-picking. If the assets were chosen to represent liquidity tiers, please state the exact sampling rule. If they are fixed benchmark assets, please name them and justify the choice.

The paper states that distributions are mean-variance normalized before computing W1. This improves comparability across quantities, but it can also hide failures in scale, variance, or dispersion. Please report both raw W1 and normalized W1. KS and W1 distances should be accompanied by confidence intervals or bootstrap intervals, especially given the small number of evaluated assets.
Temper the claims. Some claims, such as broad zero-shot geographic generalization and practical use for trading agents or stress testing, are currently stronger than what the experiments support. These should either be backed by additional experiments or stated more cautiously.

**Justification of Score**

I would assign this paper a borderline score. I appreciate the ambition and relevance of the work, but I am not fully convinced by the current empirical validation.

The paper is relevant, ambitious, and technically interesting. The proposed representation and large-scale training setup are promising, and the problem is well aligned with the workshop. However, the current empirical evidence is not yet strong enough for the breadth of the claims. The main realism evaluation is conducted on only 9 assets, the APAC evaluation is mostly perplexity-based, practical downstream usefulness is not demonstrated, important ablations are missing, and several aspects of the data and simulator are ambiguous.

I would be more positive if the authors expanded the evaluation across a larger asset set, reported uncertainty intervals, clarified the data schema, added stronger baselines, and included at least one downstream practical evaluation. As it stands, I view the paper as a promising preliminary contribution rather than a fully convincing foundation-model study for market microstructure.